# Characterization of Silver Nanoparticles Synthesized by the Aqueous Extract of *Zanthoxylum nitidum* and Its Herbicidal Activity against *Bidens pilosa* L.

**DOI:** 10.3390/nano13101637

**Published:** 2023-05-13

**Authors:** Tianying Jiang, Jinyan Huang, Jieshi Peng, Yanhui Wang, Liangwei Du

**Affiliations:** 1College of Chemistry and Chemical Engineering, Guangxi University, Nanning 530004, China1914301026@st.gxu.edu.cn (J.P.); 2Guangxi Colleges and Universities Key Laboratory of Applied Chemistry Technology and Resource Development, Guangxi University, Nanning 530004, China; 3Guangxi Key Laboratory of Biology for Crop Diseases and Insect Pests, Plant Protection Research Institute, Guangxi Academy of Agricultural Sciences, Nanning 530007, China

**Keywords:** plant-mediated synthesis, Ag nanoparticles, herbicidal activity, nanoherbicide

## Abstract

Phytosynthesis of silver nanoparticles (Ag NPs) has been progressively acquiring attractiveness. In this study, the root of *Zanthoxylum nitidum* was used to synthesize Ag NPs, and its pre-emergence herbicidal activity was tested. The synthesized Ag NPs by the aqueous extract from *Z. nitidum* were characterized by visual inspection, ultraviolet-visible spectroscopy, dynamic light scattering (DLS), X-ray diffraction (XRD), transmission electron microscopy (TEM), and energy dispersive X-ray spectroscopy (EDX). The plant-mediated synthesis was completed within 180 min and the Ag NPs exhibited a characteristic peak at around 445 nm. The results of the DLS measurement showed that the average hydrodynamic diameter was 96 nm with a polydispersity index (PDI) of 0.232. XRD results indicated the crystalline nature of the phytogenic Ag NPs. A TEM analysis revealed that the nanoparticles were spherical with an average particle size of 17 nm. An EDX spectrum confirmed the presence of an elemental silver signal. Furthermore, the Ag NPs exhibited a herbicidal potential against the seed germination and seedling growth of *Bidens Pilosa* L. The present work indicates that Ag NPs synthesized by plant extract could have potential for the development of a new nanoherbicide for weed prevention and control.

## 1. Introduction

Over the past few decades, there has been an increasing importance on the development of metal nanoparticles such as gold, silver, platinum, copper, magnesium, and zinc because of their widespread application in multidisciplinary fields. In particular, Ag NPs are the most fascinating, and studied metal nanoparticles have attracted special attention owing to their superior properties, such as high chemical stability, good electrical conductivity, excellent catalytic efficiency, and good antimicrobial activities [1,2,3].

The green synthesis of Ag NPs has been proposed as a cost-effective, energy-efficient, facile, and environmentally friendly method with the use of nontoxic natural biogenic agents [4]. The green routes are much preferable over conventional synthesis protocols due to various limitations associated with the latter such as the toxicity of chemicals, time consumption, expenditure, harsh reaction condition, and so on. Various natural products obtained from bacteria, fungi, viruses, and plant extracts act as effective stabilizing or reducing agents during the green synthesis of Ag NPs. Among them, plant extracts provide a more suitable alternative source for the synthesis of Ag NPs due to the presence of immense varieties of active phytochemicals or secondary metabolites that act as reacting agents, their sustainable nature, and the feasibility of large-scale synthesis. Different plant parts such as the leaf, shoot, stem, root, bulb, resin, fruit, flower, bark, peel, shell, seed, nut, and whole plant from various plant sources have been reported for the synthesis of Ag NPs in previous studies [5,6,7].

The Ag NPs obtained by botanical biosynthesis provide economic viability, have larger biocompatibility, and are less toxic to the biotic systems [8]. The plant-mediated Ag NPs have various specific qualities such as antibacterial, antiviral, antifungal, anti-inflammatory, antidiabetic, anticancer, nanopesticide, catalytic, dye degradation, and metal sensing properties [4]. The potential multifunctional applications of phytogenic Ag NPs are exciting and beneficial in a variety of fields, especially in various therapeutic and agricultural applications [5,6]. The potential agricultural applications of botanical Ag NPs have been well-illustrated as nanofertilizers [9], nanoscale growth regulators [10], nanoagrochemicals for germination [11,12], and nanopesticides. They are promising nanopesticides for the control of larvae, mosquitoes, harmful insects, and phytopathogens with noteworthy larvicidal, mosquitocidal, insecticidal, and antimicrobial activities [13,14,15,16].

In the present study, Ag NPs were synthesized by *Z. nitidum*, which has medicinal value and can be used and cultivated for various purposes, and the potential herbicidal activity of the biogenic Ag NPs was carried out. To our knowledge, this is the first report on the pre-emergence herbicidal potential of medicinal plant-mediated Ag NPs.

## 2. Materials and Methods

### 2.1. Materials

Silver nitrate (AgNO_3_) was purchased from Shanghai Chemical Reagent Co., Ltd., Shanghai, China. Sodium hypochlorite (NaClO) was obtained from Kelong Chemical Reagent Co., Ltd., Chengdu, China. Powder from the root of *Z. nitidum* was acquired online from Jinfangqiancao Chinese Herbal Medicine Official Flagship Store, Taobao, China. All other reagents were analytical grade and used as received. Unless otherwise stated, deionized water was used in all of the experiments.

### 2.2. Preparation of the Aqueous Extract

Firstly, 0.5 g of fine powder was added into 50 mL deionized water in a beaker and sonicated for 30 min with the ultrasonic frequency of 40 KHz at room temperature. Then, the mixture solution was centrifuged twice at 8000 rpm for 8 min and the supernatant was collected and filtered with a 0.45 μm water-phase microporous membrane. Finally, the filtrate (aqueous extract) was collected and utilized after that to synthesize Ag NPs.

### 2.3. Synthesis of Ag NPs

To synthesize the Ag NPs, 100 μL of 0.1 mol/L AgNO_3_ stock solution was added dropwise to 10 mL of the aqueous extract while stirring for 3 h at room temperature. The control groups, including only the aqueous extract or only the AgNO_3_ solution with a concentration of 1 mmol/L, were also developed using the same reaction conditions.

### 2.4. Characterization of the Synthesized Ag NPs

All the ultraviolet-visible (UV-Vis) absorption spectra were carried out on a UV-3600Plus spectrophotometer (Shimadzu, Kyoto, Japan) and measured using a 1.0 cm light-path length cuvette at the resolution of 1 nm. The spectra were investigated at a wavelength range between 250 and 900 nm. To study the reaction kinetics, 0.2 mL of the reaction solution was withdrawn at different time intervals (0, 5, 10, 20, 40, 60, 90, 120, 180, and 240 min) during the reaction process and then diluted with 0.8 mL of deionized water. Subsequently, the UV-Vis spectra of the resulting diluents were measured.

The polydispersity index (PDI) and hydrodynamic diameter of the nanoparticle solutions were measured by the dynamic light scattering (DLS) technique using a Zetasizer Nano ZS90 (Malvern Instruments, Malvern, UK). All the measurements of the nanoparticle solutions were carried out at least 3 times with a detector at a fixed angle of 90° and a constant temperature of 25 °C.

The X-ray diffraction (XRD) measurement of the nanoparticles was performed with an X-ray diffractometer (SMARTLAB 3KW, Rigaku, Tokyo, Japan) at 40 kV and 30 mA, and the radiation used was Cu-Kα. The sample for the XRD measurement was prepared by drop-coating the solution on a clean Si (111) substrate, followed by drying at room temperature.

The morphology measurements and selected area electron diffraction (SAED) analyses of the nanoparticles were carried out on a transmission electron microscopy (TEM, FEI Talos F200s, Thermo Fisher, Waltham, MA, USA) operating at an accelerating voltage of 120 kV. Elemental analysis was performed on an energy-dispersive X-ray (EDX) spectrometer equipped with the TEM instrument. The sample was prepared by dropping the nanoparticle solution onto a carbon-coated copper grid and dried at room temperature. The particle sizes of at least 200 particles were measured by using ImageJ software, and then the diameter data were transferred into Origin software for the statistical analysis of the average size and the corresponding size distribution histogram was plotted.

### 2.5. Herbicidal Activity Assay of the Synthesized Ag NPs

In this experiment, *B. pilosa* L. was chosen to be the tested weed. The dried seeds of *B. pilosa* L. were collected on the farm of Guangxi University from September to November 2022. The seeds were first sterilized by using a 2% sodium hypochlorite solution for 20 min and then washed 3 times with sterile deionized water prior to the assay. The assay was performed by using the Petri dish filter paper method with minor modifications [17,18].

Two layers of filter paper were placed on the bottom of the inverted Petri dish (90 × 15 mm) cover, and moistened with 2 mL of the synthesized Ag nanoparticle solution and the bubbles were removed. Thirty sterilized weed seeds were placed in a Petri dish with the arrangement of 5 × 6, then 3 mL of the nanoparticle solution was added, and finally, the dish was covered by another Petri dish lid. At the same time, deionized water and aqueous extract with the same volume were set up as a blank control and a solvent control, and each treatment was repeated 3 times. After treatment, all Petri dishes were transferred to a light incubator (GXZ-436D, Ningbo Jiangnan Instrument Factory, Ningbo, China) (temperature: 25 ± 2 °C, illumination intensity: 80%, light/dark: 14/10 h) for cultivation. During the cultivation process, an appropriate amount of deionized water was added to maintain the water required for seed growth. The condition of seed germination was observed every day, and the number of seed germination was recorded for 7 days. After 7 days, the germination number of seeds was counted, the root length and shoot length of germinated seeds were measured, and the inhibition rates were calculated based on the following equation:Inhibition rate (%) = (1 − *T*/*C*) × 100(1)
in which, *T* and *C* were the measured parameters of the treated seedlings and the blank control seedlings, respectively.

## 3. Results and Discussion

### 3.1. Visual Inspection of the Reaction Solution

During the synthesis of Ag NPs by plant extracts, the extraction solvents used in the preparation of plant extracts usually included deionized water, methanol, and ethanol in previous reports [6,19,20]. In this study, a water-based extraction procedure was employed for the preparation of plant extract from dried plant tissues. Water extraction could prevent the extraction of fats and lipids and obtain water-soluble bioactive components. As illustrated in Figure 1A, the obtained aqueous extract appeared pale-yellow color. The complete aqueous condition was directly provided for the synthesis of Ag NPs. Compared with the extraction at elevated or boiled temperatures, the extraction process performed at room temperature could avoid the loss of heat-sensitive components [21].

The synthesis of Ag NPs by aqueous extract of *Z. nitidum* was operated in an ambient environment and the results of the visual inspection are illustrated in Figure 1. Slightly white turbidity immediately occurred in the pale-yellow aqueous extract (Figure 1A) after the addition of the AgNO_3_ solution. As the reaction proceeded, the reaction solution became transparent and gradually deepened into reddish-brown (Figure 1B), which was the characteristic color of the Ag nanoparticle solution and indicated the formation of Ag NPs [22]. At the same time, there was no obvious color change in the control solutions. Compared with the synthesis assisted by microwave irradiation [11], elevated temperature [23], or light radiation [24], plant-mediated synthesis in this work is a greener and more effective method.

### 3.2. Reaction Process Recorded by UV-Vis Spectra

Figure 2A reveals the changes in the UV-Vis absorption spectra as a function of the reaction time during the reaction process. Curve a shows the absorption spectrum of the aqueous extract. Two small peaks centered at ca. 282 nm and 320 nm were observed, which might be derived from the biochemical constituents present in the aqueous extract. After the reaction with the AgNO_3_ solution (curves b–j), a new single absorption peak around 445 nm appeared in the spectra. This single peak is the characteristic surface plasmon resonance (SPR) of the Ag NPs, which suggests the formation of the spherical Ag NPs in the reaction solution [25,26]. The SPR values of plant-synthesized Ag NPs published in previous reports are located in the range of 400–450 nm and the difference in this value could be attributed to the biochemical constituents of the extract serving as reducing and stabilizing agents for the synthesis of Ag NPs [27]. The absorption intensity of the characteristic peaks gradually increased as the reaction proceeded, suggesting increasing numbers of the formed Ag NPs [4]. After the reaction solution was reacted for 180 min (curve i), the intensity of the absorption peak did not obviously increase when the reaction time was prolonged to 240 min (curve j). During the process of synthesis reaction, the maximum absorption wavelength of SPR peaks exhibited a slight red shift, which might be due to the continuous growth and increased size of Ag NPs [25].

Figure 2B shows a plot of the peak intensity of the absorption spectra versus reaction time. During the initial 60 min, the intensity remarkably increased and was then followed by a slow rise. Finally, the plot reached a platform after 180 min and the intensity at 240 min almost equaled that at 180 min. This plot indicates that the biosynthesis of Ag NPs by the aqueous extract was completed within 180 min. The reaction time observed in this study was relatively short compared with the previously reported times for the plant-mediated synthesis of Ag NPs [28,29].

### 3.3. Size Measurement by DLS

The average hydrodynamic diameter and particle size distribution defined using the polydispersity index (PDI) of the nanoparticle solutions were measured by the DLS technique, which was most suitable to characterize Ag NPs synthesized using plant extracts that contained various plant metabolites [6]. Considering that the size obtained from the DLS number showed a good approximation to the TEM results, the average particle size from the DLS number was selected and the value was 96 ± 21 nm [30]. The particle size distribution diagram of Ag NPs is shown in Figure 3. The PDI value was 0.232 ± 0.015, suggesting that the nanoparticle solution is a relatively monodisperse system [31].

### 3.4. Crystal Structure Measured by XRD

The XRD pattern of the synthesized Ag NPs is shown in Figure 4. Four characteristic diffraction peaks at 2θ values of 38.0°, 44.2°, 64.4°, and 77.6°, which corresponded to the (111), (200), (220), and (311) Bragg reflections of the faced-centered cubic structure of silver (JCPDS No. 04-0783), respectively [27,29]. These characteristic diffraction peaks clearly indicate the crystalline nature of the synthesized Ag NPs and the result is consistent with previous reports for Ag nanocrystals [32,33]. In addition, two weak diffraction peaks at about 32.2° and 46.1° (marked with a star) were observed in the pattern, which is common in the reported XRD patterns of Ag NPs [1,19,23,34,35]. It could be associated with the reflections of Ag_2_O left in small quantities after the reaction [36,37].

### 3.5. Morphology Measurement by TEM

TEM was used to characterize the shape and size of the synthesized Ag NPs. Figure 5A displays the representative TEM image. It can be seen that the Ag NPs were almost spherical. Figure 5B shows the corresponding size distribution histogram determined from the TEM images. As observed from Figure 5B, the synthesized nanoparticles were mainly distributed in the range of 10–22 nm with an average particle size of 17 nm by the digital analysis of TEM images containing at least 200 particles. It was noted that the average particle size estimated from the TEM measurements was smaller than that obtained from the DLS measurements, which could contribute to the aggregation of nanoparticles in the solution, the presence of hydrated capping agents such as biomacromolecules, and solvation effects during the DLS analysis [38,39]. The high-resolution transmission electron microscopy (HRTEM) image in Figure 5C illustrates a clear lattice fringe with a spacing of 0.238 nm, which corresponds to the (111) plane of silver [40]. The SAED result in Figure 5D shows concentric rings with intermittent bright dots and corresponds to (111), (200), (220), (311), and (420) planes from the inner ring to the outer ring, respectively. The results indicate the polycrystalline structures of the synthesized Ag NPs, which also corroborated our XRD measurement.

### 3.6. Elemental Analysis by EDX

The presence of elemental silver in the synthesized Ag NPs was confirmed by EDX analysis. The results are illustrated in Figure 6. The strong bands peaking at about 3 keV corresponding to the typical signal of metallic silver nanocrystallite show the reduction of silver ions to elemental silver [40,41,42]. The strong C and Cu signals might be due to the carbon-coated copper grid used in the preparation of the sample [22]. The presence of the Cl element at ca. 2.6 keV could be from the aqueous extract, which was associated with the turbidity of the reaction solution upon the addition of the AgNO_3_ solution to the extract. The weak signals of O, Mg, and Si elements could arise from the phytochemical compounds in the aqueous extract that were near or bound to the surface of the nanoparticles [43].

### 3.7. Evaluation of Pre-Emergence Herbicidal Activity

The herbicidal activity of the synthesized Ag NPs on the seed germination of *B. pilosa* was investigated and the cultivation results after 7 days are illustrated in Figure 7. During the seed growth process, the inhibitory effect of the synthesized Ag NPs on the germination of *B. pilosa* was observed and recorded. After cultivation for 3 days, the germination rates were 52.22%, 37.78%, and 14.44% for the treatments of water, aqueous extract, and the synthesized Ag NPs, respectively. After cultivation for 7 days, the inhibition rates on seed germination of 11.86% and 18.64% were obtained for aqueous extract and Ag NPs compared with the water treatment. It was obvious that the synthesized Ag nanoparticles could inhibit and delay the seed germination of *B. pilosa*.

The herbicidal activity on the seedling growth (root and shoot elongation) of *B. pilosa* was also investigated and the results of root and shoot lengths are illustrated in Figure 8. Compared with the water treatment, the aqueous extract showed a slight inhibitory activity against the seedling growth of *B. pilosa*. However, the synthesized Ag nanoparticle solution had an obvious effect on the root and shoot lengths of *B. pilosa* with inhibition rates of 19.38% and 23.33%, respectively.

As suggested above, the aqueous extract of *Z. nitidum* had a slight phytotoxic effect on *B. pilosa*. According to a previous report, it is possible that plant extracts could inhibit weed germination and seedling growth [44]. Furthermore, Ag NPs synthesized by the aqueous extract exhibited nanoherbicidal potential against the seed germination and seedling growth of *B. pilosa*. The study showed that plant-synthesized Ag nanoparticles could be a suitable alternative source for nanoherbicide formulation.

## 4. Conclusions

In this study, the environmentally friendly and cost-effective procedure was presented for the synthesis of Ag NPs by exploiting renewable natural resources. The aqueous extract of *Z. nitidum* was successfully used for the green synthesis of Ag NPs at room temperature within 180 min. The characterization of the obtained Ag NPs was completed by several analytical methods. Its herbicidal activity was evaluated by using *B. Pilosa* as the tested weed. This study showed a green synthesis of Ag NPs with promising pre-emergence herbicidal potential.

## Figures and Tables

**Figure 1 nanomaterials-13-01637-f001:**
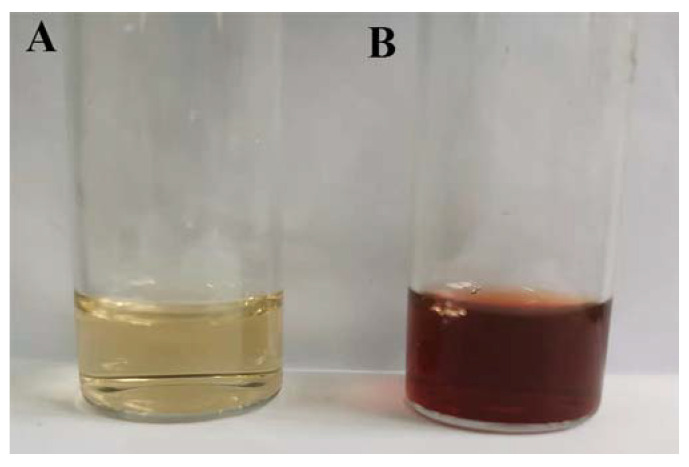
The color photographs of the aqueous extract (**A**) and after reaction with AgNO_3_ (**B**).

**Figure 2 nanomaterials-13-01637-f002:**
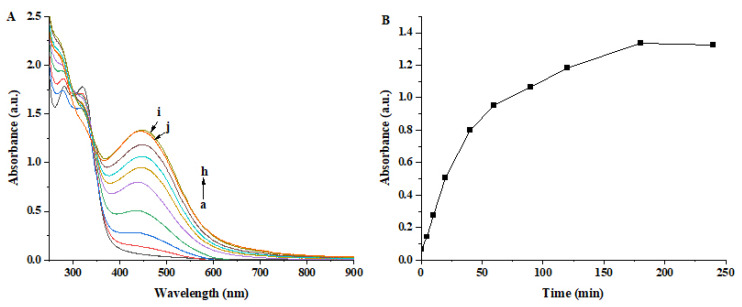
(**A**) The absorption spectra of the synthesized Ag NPs as a function of the reaction time, in which curve a was the spectrum of the aqueous extract, curve b–j corresponded to the spectra of reaction time at 5, 10, 20, 40, 60, 90, 120, 180 and 240 min, respectively. (**B**) The changes of maximum absorbance against the reaction time during the aqueous extract reacting with AgNO_3_ solution.

**Figure 3 nanomaterials-13-01637-f003:**
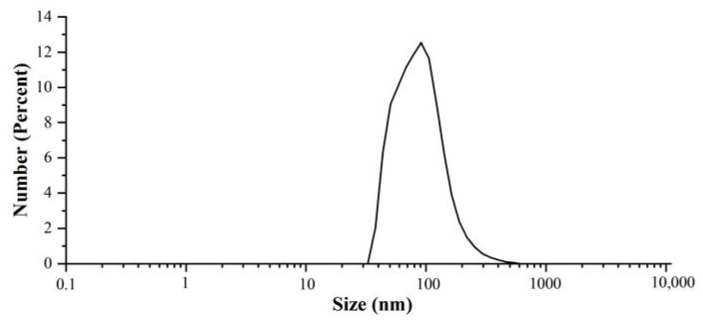
The particle size distribution diagram of the Ag NPs synthesized by aqueous extract of *Z. nitidum* measured by DLS.

**Figure 4 nanomaterials-13-01637-f004:**
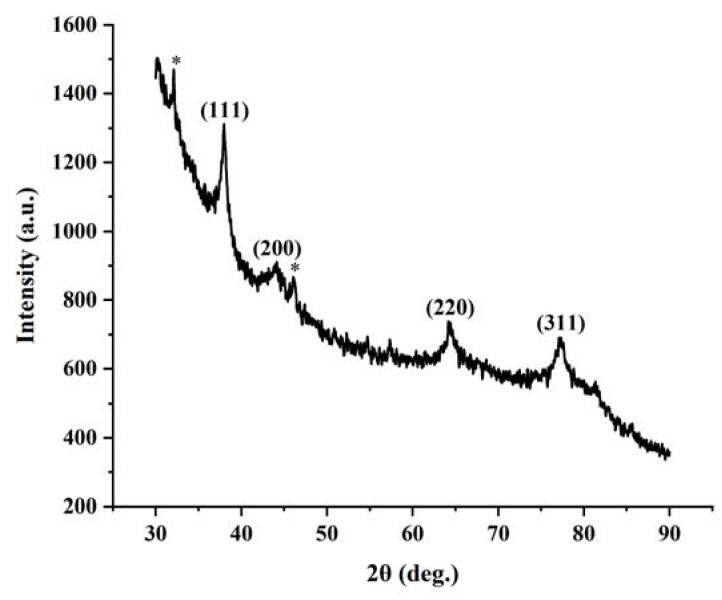
XRD pattern recorded from the powder of air-dried Ag NPs. The labelled peaks corresponded to the characteristic diffraction peaks of elemental Ag. The peaks marked with star indicated the unidentified peaks.

**Figure 5 nanomaterials-13-01637-f005:**
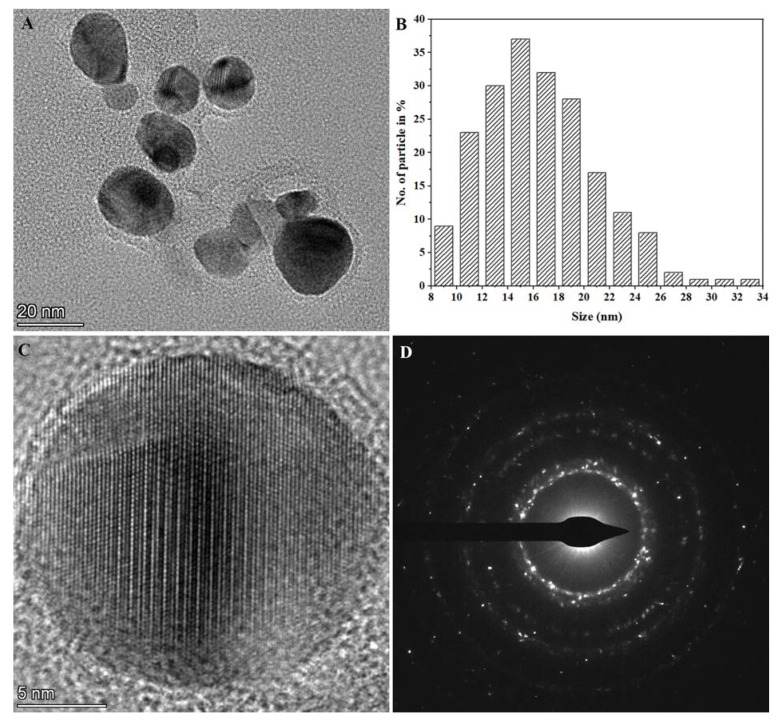
TEM analyses of the synthesized Ag NPs. (**A**) Representative TEM image, (**B**) particle size distribution histogram counted from TEM images, (**C**) typical HRTEM image, and (**D**) the SAED pattern.

**Figure 6 nanomaterials-13-01637-f006:**
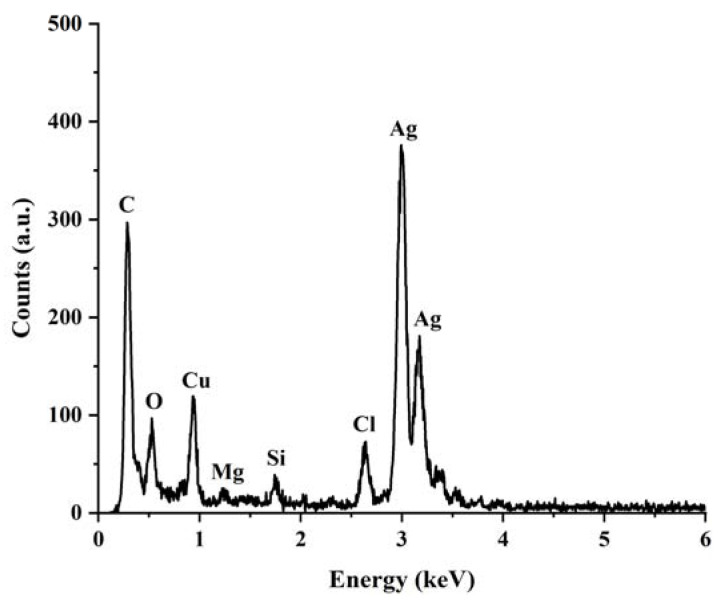
EDX spectrum of the synthesized Ag NPs. The labelled peaks corresponded to the elements present in the sample.

**Figure 7 nanomaterials-13-01637-f007:**
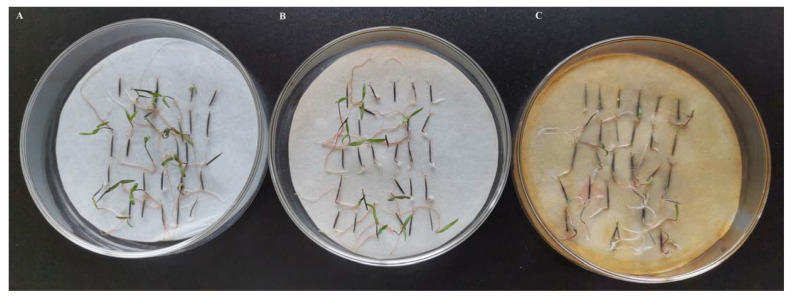
Evaluation of the herbicidal activity of (**A**) water, (**B**) aqueous extract, and (**C**) the synthesized Ag NPs against *B. pilosa* after cultivation for 7 d.

**Figure 8 nanomaterials-13-01637-f008:**
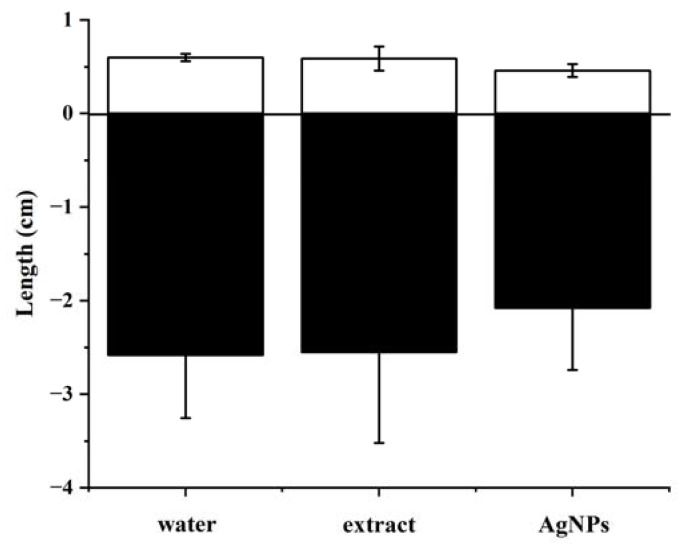
Effect of water, aqueous extract, and the synthesized Ag NPs on the shoot length (□) and root length (■) of *B. pilosa*. The data in the figure were given as mean ± SD for three repetitions.

## Data Availability

Data is contained within the article.

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
