# Peer review of "Characterization of Silver Nanoparticles Synthesized by the Aqueous Extract of Zanthoxylum nitidum and Its Herbicidal Activity against Bidens pilosa L."

_nanomaterials, 2023, doi:10.3390/nano13101637_

Round 1

Reviewer 1 Report

This work presents a preparation method for the production of Ag nanoparticles (NPs) based on using Zanthoxylum nitidum as “bio”-reductant. Subsequently, the herbicidal activity of the as-prepared NPs was also exhibited.  While the concept is not a breakthrough, the conduct of the study and the results of the herbicidal tests made the manuscript interesting enough to be published. However, the manuscript has many weaknesses that need to be improved before publication. These are detailed as follows:

1, It needs to be clarified, how it is possible that the hydrodynamic radius (223.2 nm) and the average particle size (16.6 nm) differ from each other at this high level? This difference suggests that DLS-intensities were used for the calculation. But this concept is wrong because DLS-numbers must be applied for a comparison with TEM measurements. To revise this section, use this article: T. G. F. Souza et al. “A comparison of TEM and DLS methods to characterize size distribution of ceramic nanoparticles” (2016) J. Phys. Conf. Ser. 733 012039. DOI: 10.1088/1742-6596/733/1/012039.

Furthermore, DLS spectra need be added to the manuscript to be more precise and useful.

2, In the XRD diffractogram, the star-marked reflections of “impurities” are more likely to be associated with the reflections of Ag2O left in small quantity after the reaction. To clarify this section, use this article: F. John and G. Madhumitha “Biomolecules Derived from Carissa edulis for the Microwave Assisted Synthesis of Ag2O Nanoparticles: A Study Against S. incertulas, C. medinalis and S. mauritia” J. Cluster Sci. (2019) 30 1243. DOI: 10.1007/s10876-019-01627-3.

3, It is practically impossible for the copper-content to appear in the EDX spectrum if it is in the sample holder. This section needs to be revised and it is important to confirm that the extract is free of copper. Otherwise, some or all of the measured herbicidal activity could be related to the copper content.

4, The introduction states that “…Ag NPs as the most fascinating and studied metal nanoparticles have attracted special attention owing to their superior properties, such as high chemical stability, good electrical conductivity, excellent catalytic efficiency, and well-showed antimicrobial activities [1,2].” Among the references, however, there is one example of its antimicrobial activity. Examples should be given for the other areas. To present its catalytic activities, the next article must be consulted: R. Meszaros et al. “Exploiting a silver–bismuth hybrid material as heterogeneous noble metal catalyst for decarboxylations and decarboxylative deuterations of carboxylic acids under batch and continuous flow conditions” Green Chem. 2021 23 4685. DOI: 10.1039/D1GC00924A.

Considering my above-mentioned concerns and criticisms, I recommend this manuscript for publication in the MDPI journal “Nanomaterials” after major revision.

Reviewer 2 Report

The paper reports about the preparation and characterization of silver nanoparticles by means extract of a plant. The use of natural extracts as reducing agents to prepare silver NPs is the object of many published works and the present study does not add any new finding to justify the publication. On top of that the characterization procedure (as well as the herbicidal tests) has many drawbacks as detailed below. Hence, the paper cannot be accepted for publication.

1) Page 2, lines 60-63. Besides the poor english, the claim “…herbicidal potential of the medicinal plant-mediated Ag-NPS” is not supported by the results.

2) Page 4, line 175-180. The plot of the peak at ca 450 nm vs reaction time shows a sudden increase in the first 5 minutes. Therefore, measurements at shorter time (every 30 sec. for instance) had to be performed in the first 5 minute of the reaction.

3) Page 5, line 192. I guess that the size reported is the “Z-average” value of the hydrodynamic diameter. A reasonable accuracy for the size is +/- 1 (if not 10) nm. +/- 0.2 nm is non sense.

4) Page 5, line 203.  The (111) peak of silver is the most intense due to the structure of the metal (check the JCPD card of silver). It is not an evidence for the (111) preferential orientation of the Ag NPs. Moreover, this preferential orientation would not be consistent with the almost spherical shape (page 6, line 212) of the NPs observed in the TEM images.

5) Page 7, lines 232-234. “Surface plasmon resonance”???? These are not absorption peaks but peaks due to the emission of x-rays. The peak at ca. 2.6 keV attributed to Ag is probably due to chlorine.  This would suggest the presence of silver chloride in agreement with the observation of the turbidity of the solution upon addition of the AgNO3 solution to the extract (page 4, line 146).

6) Page 8. The evaluation of the herbicidal activity is not convincing.

English must be revised

Round 2

Reviewer 1 Report

The authors asked all my questions properly. Thus, I suggest this manuscript for acceptance in this present form.

Author Response

Thanks for your valuable suggestions, which help us a lot to improve this paper.

Reviewer 2 Report

The authors only in part answer to the questions arose in the previous review. The following points must be addressed before the paper could be accepted for publication.

1) Page 5, lines 195-197 and line 20 of abstract. A reasonable value of the accuracy of the hydrodynamic diameter determined by DLS is +/- 1 nm. Looking at the curve in figure 3 a more conservative uncertainty would be +/- 10 nm. So the size should be reported as 96 +/- 10 nm. 96.1 +/- 0.2 nm is misleading. Also the size determined by TEM (page 7, line 226)  should be reported as 17 nm (not as 16.6 nm).

2) Figure 4. The authors did not explain the meaning of the stars close to some peaks in the diffractogram.  

3) Elemental analysis by EDX. The peak at ca. 2.6 keV is not due to silver but, probably, to chlorine. The only peaks due to Ag are those at ca. 3.0 keV (L alpha1,2), 3.1 keV (L beta1) and 3.4 keV (L beta2).  See for instance https://xdb.lbl.gov.

After addressing the points reported the paper can be accepted for publication
